# Advances in Non-Invasive Neuromodulation: Designing Closed-Loop Devices for Respiratory-Controlled Transcutaneous Vagus Nerve Stimulation

**DOI:** 10.3390/healthcare12010031

**Published:** 2023-12-22

**Authors:** Gabriella Maria de Faria, Eugênia Gonzales Lopes, Eleonora Tobaldini, Nicola Montano, Tatiana Sousa Cunha, Karina Rabello Casali, Henrique Alves de Amorim

**Affiliations:** 1Institute of Science and Technology, Universidade Federal de São Paulo, São José dos Campos 12231-280, Brazil; gmfaria@unifesp.br (G.M.d.F.); eugenia.lopes@geb.inatel.br (E.G.L.); ts.cunha@unifesp.br (T.S.C.); henrique.amorim@unifesp.br (H.A.d.A.); 2Department of Clinical Sciences and Community Health, Università degli Studi di Milano, 20122 Milan, Italy; eleonora.tobaldini@unimi.it (E.T.); nicola.montano@unimi.it (N.M.)

**Keywords:** neuromodulation, tVNS, closed-loop, Node-RED

## Abstract

Studies suggest non-invasive transcutaneous auricular vagus nerve stimulation (taVNS) as a potential therapeutic option for various pathological conditions, such as epilepsy and depression. Exhalation-controlled taVNS, which synchronizes stimulation with internal body rhythms, holds promise for enhanced neuromodulation, but there is no closed-loop system in the literature capable of performing such integration in real time. In this context, the objective was to develop real-time signal processing techniques and an integrated closed-loop device with sensors to acquire physiological data. After a conditioning stage, the signal is processed and delivers synchronized electrical stimulation during the patient’s expiratory phase. Additional modules were designed for processing, software-controlled selectors, remote and autonomous operation, improved analysis, and graphical visualization. The signal processing method effectively extracted respiratory cycles and successfully attenuated signal noise. Heart rate variability was assessed in real time, using linear statistical evaluation. The prototype feedback stimulator device was physically constructed. Respiratory peak detection achieved an accuracy of 90%, and the real-time processing resulted in a small delay of up to 150 ms in the detection of the expiratory phase. Thus, preliminary results show promising accuracy, indicating the need for additional tests to optimize real-time processing and the application of the prototype in clinical studies.

## 1. Introduction

Transcutaneous auricular vagus nerve stimulation (taVNS) is a non-invasive neuromodulation therapy that eliminates the need for internal device implantation, presenting as a favorable therapeutic option. This method relies on electrical surface stimulation, bypassing the need for surgical procedures [1]. In the context of evaluating autonomic state regulation, heart rate variability (HRV) calculation stands as a commonly employed method, allowing for the assessment of sympathetic or parasympathetic system activation [2]. Consequently, evidence suggests that taVNS effectively modulates the auricular vagus nerve’s parasympathetic pathway, with studies proposing its systemic effects [3]. Non-invasive stimulation of auricular afferent receptors is achieved by employing surface electrodes on the outer ear, utilizing an electronic device to produce pulses at specific stimulation frequencies [4,5]. Despite the increased application of transcutaneous stimulation techniques, commercial devices currently operate as open-loop approaches, lacking adaptive adjustments based on the user’s physiological variables.

### 1.1. Closed-Loop Stimulation

In recent years, there has been an increasing focus on enhancing therapy by synchronizing stimulation parameters with a patient’s physiological response. This has led to the proposal of closed-loop models aiming to optimize treatment outcomes [6].

Differing from the traditional approach, closed-loop VNS allows for adjusting stimulation to the patient’s physiological conditions in real time. In this system, stimulation is automatically adjusted based on the detection of specific physiological signals or biomarkers, providing a personalized and effective intervention. In therapies for stroke rehabilitation, for instance, closed-loop VNS enables precise control over when stimulation is delivered to the patient, ensuring it occurs during appropriate moments [7], maximizing functional recovery and enhancing rehabilitation efficacy [8]. Stimulation can be synchronized with specific cardiac parameters, such as heart rate or blood pressure, to aid in the treatment of cardiovascular diseases like hypertension and myocardial ischemia [9].

Decoding techniques enabling the identification and classification of patterns assist in determining the precise timing of VNS application in cardiovascular treatments, based on the evaluation of neural activity [10]. As vagal neural activity synchronizes with respiratory and cardiac cycles, these physiological signals provide an avenue for external modulation to enhance stimulation efficacy. Furthermore, indirect and non-invasive measures like heart rate variability, baroreflex sensitivity, and respiratory sinus arrhythmia can serve as indicators of cardiac vagal nerve activity, allowing for adjustment of parameters in the stimulating device [11]. Pulmonary respiratory reflexes, facilitated by mechanical lung-stretch receptors, convey information regarding the solitary tract nucleus, thereby influencing cardiac and autonomic rhythms. In this context, by synchronizing the pulse train of stimulation with the respiratory rhythm, especially during the expiration phase, the intended neuromodulation may have a more significant physiological impact. Moreover, the acquisition and analysis of various physiological signals in response to taVNS usage can complete the therapeutic cycle, allowing for real-time assessment of the therapy, thereby facilitating more efficient optimization [3].

Advancements in wearable sensor technologies and real-time signal processing methodologies collectively form the foundation for research involving closed-loop taVNS. Not only does this approach offer innovative and personalized treatment options for the patient, but it also enables the tailoring of stimulation dosage according to the measured outcomes during stimulation, potentially reducing side effects and enhancing the overall quality of life for patients.

### 1.2. Sensors for Signal Acquisition

Advancements in sensor technology have enabled the precise and continuous collection of physiological data, such as heart rate, blood pressure, stress levels, and even brain activity, in an unobtrusive and real-time manner. These data are crucial for evaluating the effects of therapy and tailoring treatments to meet the specific needs of each patient. Commercial devices, such as smartwatches, are commonly used by athletes to track daily vital signs and performance during activities throughout the day. They are also employed for the continuous monitoring of patients, especially the elderly, in a discreet and comfortable manner, without exposing that the individual is under care. More sophisticated devices, like biosensors and wearable sensors [6], allow for the recording of variables used to monitor blood glucose levels, oxygen saturation, heart rate, and blood pressure [12].

### 1.3. Methodologies for Signal Processing and Real-Time Variable Detection

Methods for extracting physiological data using real-time signal processing are currently utilized in both research and medical devices to support remote treatment [13]. Although tools for categorizing time-varying signals are widely adopted in diverse research areas, the majority of these solutions depend on algorithms crafted for high computational intensity. This poses challenges when implementing them in small embedded systems that have limited resources [14].

In stimulation therapies, the assessment of heartbeats can assist in the safe practice of therapy. Through real-time signal processing methodologies, instantaneous detection of a drop in heart rate to critical levels, due to vagal stimulation, can indicate to the user to adjust or interrupt the session. In the context of autonomic assessment, heart rate variability (HRV) is widely used to evaluate these systems’ activation [2]. Among the challenges associated with this metric, the need for wider time windows for frequency domain analysis stands out, as well as the requirement for high-sampling-rate sensors to detect pulse intervals and the capacity of the processing unit remotely.

Quantitatively, changes in heart rhythm patterns can be obtained with beat recognition algorithms, such as QRS complex detection. Temporal domain analyses can be performed using calculations involving “mean RR, SDNN, rMSSD, pNN50”. In turn, signal transformation methods are employed to analyze HRV indices in the frequency domain, such as fast Fourier transform (FFT), discrete Fourier transform (DFT), wavelet transform, or auto-regressive (AR) modeling [15]. Developed in collaboration between the European Society of Cardiology and the North American Society for Pacing and Electrophysiology, these methodologies delineate protocols and clarify correlations among physiological assessment parameters [16].

While frequency domain methods are effective for assessing biological variables not observable in the time domain, they necessitate a longer signal window—typically at least two minutes—to extract the LF, HF, and VLF bands. Obtaining other parameters may require even lengthier measurements [17]. This requirement could limit their utility in short sampling periods, particularly in real-time event identification.

Conversely, methods for processing respiratory signals are commonly observed in respiratory performance tests and polysomnography examinations. Some studies have proposed algorithms that utilize tools like discrete wavelet transform (DWT) and FFT for sleep respiration detection [18]. The accurate detection of the end of inspiration and the start of expiration has been investigated by other researchers as well [14,19]. In order to synchronize the moment of stimulation triggering and to optimize the effects of vagal modulation through transcutaneous stimulation, our research team developed a mathematical model [20]. The online processing steps applied to the recorded respiratory signal highlight the complexity of managing signal variations and irregular patterns [21].

### 1.4. Remote Therapies

The implementation of advanced signal processing techniques enables precise and controlled delivery of therapies in conjunction with a parallel assessment of effects. Therefore, stimulation devices can be combined with monitoring systems to evaluate patient responses to electrical stimuli. This integration allows for better control of therapeutic interventions and the ability to adjust stimulation settings based on the specific needs of the patient [22].

The advancement of remote technologies has made it possible to offer remote therapies, surpassing geographical barriers and granting access to specialized treatments, even for patients in remote areas. Telemedicine and digital health platforms facilitate remote patient monitoring, treatment adjustments, and real-time interactions with healthcare professionals. This fosters a more convenient and flexible approach while enhancing treatment adherence [13]. These advancements suggest a reconfiguration of healthcare management, aiming to prioritize preventive care and well-being, shifting away from the traditional focus solely on crisis management and disease treatment in conventional healthcare systems. The advancement of continuous and remote monitoring is achieved through the application of real-time data processing, transmission, and integration with cloud-connected sensors and devices [23,24]. The process involves the segmentation of data into transmission packets, wherein each packet contains identifiable information that is utilized for subsequent processing, aiming to accurately extract parameters.

In this study, our primary objectives revolve around innovating closed-loop taVNS therapies by synchronizing stimulation parameters with respiratory rhythms. The aim is to optimize the therapy’s physiological impact by leveraging real-time signal processing and advanced sensor technologies. Specifically, our focus includes developing a closed-loop system capable of real-time synchronization with respiratory rhythms to enhance the effectiveness of taVNS. Furthermore, we aim to prototype and validate a remote and autonomous device that integrates signal conditioning, respiratory signal processing, and precise stimulation timing, thus enabling safe and effective therapy with improved patient outcomes.

## 2. Materials and Methods

This present study involves the development and assessment of a real-time exhalation detection algorithm synchronized with the pulse train at a user-defined frequency for taVNS. The mathematical model records, filters, and identifies the expiration phase of the respiratory signal through an automated real-time model. The output of the processing model comprises a pulsating signal (pulse train) whenever the expiration phase is initiated. This pulsating output triggers the implemented electrical circuit that generates single-phase pulses for stimulation, targeting the ear region with the predefined frequency and voltage designated for the session. All synchronization and parameter adjustments can be carried out during the therapy without affecting the code’s performance or the designed electrical pulse generator circuit.

The developed stimulator was structured into five stages: (1) signal acquisition and conditioning module, (2) implementation of the physiological signal processing respiratory algorithm in real time, (3) pulse generation circuit (PGC) with voltage regulation module and synchronization with the exhalation phase, along with encapsulation design, and (4) usability and web interface for parameter adjustments.

### 2.1. Signal Acquisition and Conditioning Module (Manufacturing)

Heartbeats are captured using the MAX 30102 photoplethysmography sensor, which incorporates a module for monitoring heart rate and pulse oximetry. Equipped with LEDs and photodetectors, the sensor detects changes in blood flow while accurately measuring the user’s pulse. The sensor works in collaboration with an MCU, and the electronic board was designed using Autodesk’s Eagle software, v9.6.2 (Figure 1A). The respiratory signal is obtained through a thermistor sensor positioned within a nasal cannula to record temperature variations during the inhalation and exhalation of air. This circuit includes passive analog filtering elements and an instrumentation amplifier. The MCU is interconnected with the board to facilitate the integration of the mathematical model, enabling the reception and digital processing of the analog signal. The initial step involved implementing the logical diagram of the circuit, ensuring connections of the same logical signal level (Figure 1B).

After establishing the logical circuit design, we proceeded to create the physical board layout, known as the footprint (Figure 2A,B). Lastly, using the same software integrated with Fusion 360, individual 3D models of each component are implemented, utilizing resources like Grabcad, which offers a library of 3D models for electronic components (Figure 2C). It is crucial to guarantee the project’s feasibility for subsequent manufacturing, considering aspects like soldering access and component placement. This enables a 3D visualization of the board before proceeding with fabrication.

With the complete board design, the footprint file was printed onto a transparent sheet, sensitized with dry film, cleaned and drilled, after corrosion process (Figure 3).

After completing the entire board design, the footprint file was printed onto a transparent sheet, treated with a sensitive dry film, cleaned thoroughly, and then drilled as part of the corrosion process (Figure 3).

### 2.2. Real-Time Implementation of the Physiological Signal Processing Respiratory Algorithm

The implementation of expiration-controlled taVNS requires the development of real-time signal processing methodologies to accurately detect the signal and synchronize the stimulus at the precise moment. A crucial aspect involves incorporating a linear regression model and subroutines for identifying the trend of the respiratory signal, as determined by the expiration phase detection algorithm (EPDA). The MCU samples the respiratory signal at a frequency of 25 Hz. The sampled signal exhibits behavior resembling a sine wave, with an increase in signal amplitude during inspiration and a gradual decline during expiration, often accompanied by high-frequency noise. To manage computational resources efficiently, a ring buffer concept was employed, ensuring the optimal use of memory and enhancing performance (Figure 4) [25]. These routines are executed sequentially for every new sampled signal point, facilitating high-resolution signal processing for real-time analysis.

The implemented routine for processing respiratory signals involves acquiring a new vector comprising the latest sampled point. This vector serves as a trigger for subsequent processing routines, including the computation of the moving average for signal filtering or performing linear regression. The flow of the algorithm presented is explained in Appendix A, along with the variables and parameters proposed for each computation. Figure 5 represents the logic stages of the implemented algorithm. After the signal is received and set up, the dynamic vector of the ring buffer is used. This means that a digital low-pass filter (moving average) is put in place to reduce noise (stage 1).

Both the raw and filtered signals are not stored in the MCU memory to reduce processing time, improve response time, and reduce transport delay. They are updated with a new sliding vector of five points for each one. This is used to determine the linear regression (stage 2) and the slope of the respiratory curve (stage 3), which finds the highest and lowest points. They are then cleaned from memory to reduce processing. Following that, an outlier removal routine detects sinusoidal patterns that differ from the expected physiological breathing pattern (stage 4).

The previously mentioned procedures, pertaining to stages 1 through 4, are illustrated in Figure 6.

During the identified expiration phase, the algorithm concurrently activates a function designed to adjust the frequency of the stimulation pulse train. This parameter is user-defined and adjustable through the therapist interface on the web portal (stages 5 and 6). This output is transmitted to the MCU output port, referred to as the pulse train signal (PTS) (stages 7 and 8). This process continues as long as the function remains active, corresponding to the real-time identification of the expiration phase. These data are synchronously transmitted from the web portal to the MCU via serial communication for respiratory signal visualization. Therefore, the processing module generates a pulsating signal (PTS) at the appropriate frequency exclusively during the exhalation phase, which is observed at the output of the microcontroller unit (MCU). The output is connected to the pulse generator circuit (PGC). Figure 7 displays the illustrations of stages 1, 3, and 7 of the EPDA.

### 2.3. HRV Analysis

Intervals between beats were computed within a circular buffer to calculate the rMSSD index. This approach enables continuous signal measurement throughout the stimulation period. Studies suggest that the extraction of data with enhanced statistical significance relies on the size of the moving buffer window. For instance, a minimum recording interval of 30 s for heartbeat recording in ECG may be necessary to extract reliable parameters [26]. Accordingly, the current model has been updated to enhance the precision of the earlier algorithm [27]. Moreover, additional cardiac signal parameters have been integrated for plotting on the web portal, including a vector of RR intervals and heart rate. To facilitate broader applicability, a normalized rMSSD value was proposed, aligning with commercial applications. While the rMSSD was typically measured in milliseconds, the baseline value and its variation, which increase with parasympathetic activation, may vary across users [28]. Hence, the suggested model normalizes the identified index within a range of 1 to 100, facilitating a comprehensive evaluation of the therapy’s efficacy.

### 2.4. Pulse Generation Circuit (PGC) Incorporating Voltage Regulation and Synchronization with the Exhalation Phase, along with Encapsulation Design

Two printed circuit boards were developed for the manufacture of the PGC and the stimulation voltage regulation module. A voltage doubler module was implemented to increase the voltage supplied from the PGC up to four times its original value. Following the voltage boost, a voltage regulator stage within the 1.27–12 V range was incorporated, enabling users to adjust the output voltage to the desired stimulation limit. To facilitate this, a potentiometer and an LCD display were integrated into the device’s casing, enabling users to regulate and monitor the output voltage during therapy. Concurrently, the PTS output from the MCU is linked to this module. Consequently, during expiration, the PTS output generates a signal comprising high and low rectangular pulses, which is then connected to a switching circuit to generate a pulse train, producing a variable pulse train with the user-regulated voltage, subsequently linked to the electrodes intended for placement on the auricular branch.

The design of the stimulation voltage regulation module (Figure 8A) and pulse generation circuit (PGC) (Figure 8B) was represented through logical diagrams. The board project in Eagle software was designed for both modules (Figure 8C), as explained before in the topic “Signal acquisition and conditioning module board”.

### 2.5. Usability and Web Interface for Parameter Adjustments

The MCU was connected to the computer via a USB cable, enabling the evaluation of the respiratory signal and the performance of the pulse train during stimulation. To facilitate this process, a web interface is currently under development using Node-RED software, v2.2.2. A dedicated patient data tab has been incorporated to enable the recording of pertinent information, including age, gender, baseline heart rate variability, stimulation duration, and the selected frequency. The interface also has frequency selectors that let users change the frequency and length of stimulation at any time during the therapy session. The frequency ranges from 2 Hz to 25 Hz. Finally, the interface provides real-time visualization of HRV for immediate monitoring when the user employs the sensor for cardiac surveillance.

After the manufacturing and assembly of all modules, validation tests were conducted for each stage of EPDA. To ensure a comprehensive assessment of the device’s functionality, precision tests for exhalation detection were performed. Six measurements, each lasting three minutes, were obtained to evaluate the precision of the system. These measurements encompassed both the unprocessed and filtered respiratory signals. The filtering process involved the application of a moving average filter with varying window sizes to enhance signal accuracy and reliability. This detailed validation approach aimed to assess the performance of individual components and their seamless integration into the overall functionality of the EPDA.

The modules detailed in Section 2.1, Section 2.2, Section 2.3, Section 2.4, Section 2.5 underwent rigorous unit testing procedures aimed at ensuring the utmost code quality. This meticulous testing process involved in-depth scrutiny of each module to detect and resolve potential issues. All subsystems underwent meticulous testing procedures, both in isolation and when integrated, to guarantee flawless operation. To evaluate the effectiveness of the developed device, a cohort of six healthy individuals, free from any comorbidities, was selected for this study. The participants, with an average age of 40 ± 1 years (2 male and 4 female), were instructed to maintain a seated position and follow regular breathing patterns. To accurately capture respiratory signals, the thermistor sensor was strategically placed near the nasal region of each participant. Device activation was initiated, and a stimulation frequency of 25 Hz was chosen via the user interface. This frequency selection aimed to conduct a comprehensive and rigorous assessment of the equipment, ensuring its effective operation under diverse conditions.

To assess hardware limitations and signal processing in embedded systems, we conducted stimulation sessions that incorporated a comprehensive feedback circuit. During these sessions, we observed the behavior of the respiratory signal, examined linear regression patterns, and studied the switching of the pulse train at various frequencies. This evaluation was crucial, considering the inherent challenges associated with the effective and accurate implementation of detection algorithms in real-time scenarios. This is particularly relevant in applications involving instantaneous biofeedback and respiratory synchronization [14,29], where the complexity of the algorithms can impact their real-world performance.

## 3. Results and Discussion

In this section, we present the results obtained from our study, which focused on the effective use of real-time signal processing techniques along with an integrated vagal stimulation device, developed within a closed-loop system. In order to achieve complete validation of each component implemented, this topic additionally analyzes the outcomes achieved through a series of signal processing procedures. These steps involve the application of filtration and amplification techniques to the user’s respiratory signals, followed by their transmission to an embedded microcontroller. This microcontroller carries a real-time exhalation detection algorithm that is coordinated with user-defined frequencies for auricular vagal stimulation. The resulting output consists of a square signal that generates a pulse train composed of a single-phase pulse, precisely timed and designed to stimulate the ear region at predetermined frequencies and voltages. In addition, the device prototyping involved the creation of printed circuit boards that incorporated analog circuitry and MCU. Furthermore, we present advancements in graphical visualization tools specifically designed for respiratory and cardiac signals, along with preliminary studies with users that validate the accuracy and sensitivity of our prototyped device.

### 3.1. Signal Acquisition and Conditioning Module (Manufacturing)

The thermistor, utilized for the acquisition of the respiratory signal, was carefully positioned near the user’s nostrils with the assistance of a microphone accessory (Figure 9A). The PCB board, developed following the outlined methodology, was adept at accommodating all the components of the signal conditioning circuit, as well as the MCU (Figure 9B).

### 3.2. Evaluation of Real-Time Physiological Signal Processing Algorithm

As depicted in Figure 4 of the methodology section, the EDPA algorithm utilizes a dynamic window for conducting real-time analysis and extracting ascending and descending signals. This procedure entails the continuous acquisition of new data while discarding older data from memory using a series of sliding windows. To visually showcase the effectiveness of the EPDA algorithm in detecting the expiration phase in real time, data were initially recorded in a .csv file and subsequently cleared during each processing iteration. The subsequent sections illustrate the results of each phase, as described in the EDPA algorithm (Figure 5).

Stage 1—Move average filter: the respiratory signal underwent real-time digital filtering using window sizes of N = 5 and N = 10 points. The N = 10 window filter demonstrates improved noise reduction but also introduces a longer delay in the filtered signal. Figure 10A illustrates the raw respiratory signal (in blue) and the N = 10 filtered signal (in red).

Stage 2—Linear Regression: at each of the five sampled points, the result of linear regression produced a data vector consisting of the angular coefficients of the straight line connecting these points.

Stages 3 and 4: Figure 10A also depicts the rectangular pulse signal (in orange) that demarcates the inspiration (low level) and expiration (high level) phases (expiration phase detection), along with the filtered signal. The resulting square wave oscillates according to the slope of the respiratory signal, and it is this state that switches the next stages. Through the process of outlier removal, intervals with strong signals that do not match the physiological respiratory cycle are also taken out of the analysis.

Stages 5 and 6: following the detection of the initiation of the expiration phase, the phases comprising the serial port reading and frequency parameters check of the EPDA algorithm watch the stimulation frequency selected in the user’s web interface and adjust it accordingly to reach the specified period.

Stages 7 and 8: the algorithm modifies the state of the PTS output based on the selected frequency. To illustrate the outcome of this particular stage, Figure 10B displays the PTS signal corresponding to a specific frequency of 20 Hz, depicted alongside the breathing signal and square wave.

The evolution of the angle derived from the linear regression line, as illustrated in Figure 11, was depicted for specific signal segments ranging from region 1 to region 16. Accuracy tests revealed that a vector consisting of five points was adequate for determining the slope of the line and proved applicable to all respiratory signals collected. This precision is crucial for identifying the inflection points of the signal, which, in turn, correspond to the onsets of the inspiration and expiration phases.

A second analysis of the results is based on evaluating the period of inspiration and expiration for plotting and statistical analysis using the signals presented in Figure 12 to explain the performance results. Square-wave and low-to-high signal transition times were recorded.

Statistical analyses were conducted using the MCU-filtered signal to assess the precision and accuracy of the EPDA model. In Figure 12A, we present the filtered signal alongside the rectangular EPDA output pulse. Figure 12B displays points R1 (representing the measured start of the expiration phase) and E1 (indicating the moment identified by EPDA), while points R2 and E2 represent the measured and EPDA-identified starts of the expiration phase, respectively. The delay time taken by our algorithm to detect the initiation of expiration was calculated by measuring the time interval between the peak identified using the ‘findpeaks()’ function from the Matlab software, vR2021a and point E1 obtained through real-time analysis. This analysis included a semi-automatic routine for outlier removal and false positives, as well as the measurement of the time interval (DT delay) for statistical analysis (Figure 12C).

The analysis encompassed six respiratory signals, which were real data collected during an experimental protocol involving healthy subjects. Each signal had a duration of 3 min, excluding nonstationary segments contaminated by noise originating from speech or movement artifacts. These data represent authentic physiological signals obtained directly from the study participants, providing a robust and reliable foundation for the analysis. The algorithm achieved a peak detection accuracy of over 90% with a maximum delay of 150 ms. Figure 13 illustrates the average delay time through a boxplot analysis, emphasizing the largest deviation and providing a visual representation for the evaluation of response delays.

The MCU processes the 25 Hz stimulation frequency delivered for stimulation at an equivalent processing rate of 20 ms per cycle. The processing performances of the previously mentioned routines were evaluated through testing conducted on an oscilloscope. The statistical values obtained demonstrated high levels of accuracy and minimal latency when employed in conjunction with the moving window technique. This suggests that the performance is significantly enhanced, resulting in improved efficiency and reduced storage requirements. In subsequent investigations, alternative approaches for mitigating delay, such as segmenting the stages into distinct processing hues, may be subjected to empirical examination. Therefore, additional accuracy tests may be necessary to optimize the proposed real-time processing, and further studies are required for a comprehensive, long-term assessment of effects.

### 3.3. Pulse Generation Circuit (PGC) with Voltage Regulation and Synchronization with the Exhalation Phase and Encapsulation Design

Individual results of the PGC block used in this article were tested and validated in previous works within our research group [20,27]. The model of the pulse train generation circuit was proposed and simulated, assembled on a breadboard and validated by visualizing the frequencies of interest using an oscilloscope (Appendix B). The modules were validated by simulating the circuits using LTspice software, v17.0.30.0.

The casing for encapsulation underwent a validation process through additive manufacturing using a SethiAip printer. Following the initial print, specific refinements were identified to fine-tune its dimensions and facilitate the installation of components, as well as to ensure a proper fit for the two parts. The ultimate outcome of the casing is displayed in Figure 14, with three boards of PGC integrated. The first board contains the respiratory signal acquisition and conditioning module, the second board contains the voltage regulation module connected to the switching circuit, and the third board contains the voltage doubler circuit.

Figure 15 illustrates the behavior of the signal generated by the PGC, synchronized with the user’s breathing through data processing. Stimulation electrodes were connected to an oscilloscope for signal analysis. Notably, the ‘pulse train’ signal, active during exhalation, follows a high-level signal initiated during inhalation. The high-level signal was designed to enhance the sensitivity of the electrode placed in the auricular region, although it can be fine-tuned based on more specific testing. This pattern repeats throughout the respiratory cycle and therapy sessions. In this example, the stimulus voltage was maintained at 9.2 V (regulated by the potentiometer), and the user-defined pulse train frequency was set to 25 Hz.

### 3.4. Usability and Web Interface for Parameter Adjustments

The assembled stimulator, equipped with integrated algorithms within the MiniEsp 32, simplifies the functioning of the stimulation device. During usability tests, thermistor sensors are employed to capture the respiratory signal, while ear electrodes are utilized for administering the stimulation. The positioning of these peripherals in relation to the user is depicted in the figures below, along with the connections of these peripherals to the stimulator (Figure 16).

The implemented web interface is depicted in Figure 17, emphasizing the patient data collection form and the stimulation frequency selectors. It is important to note that, in the experimental protocol with healthy subjects, all participants utilized the device. The signal visualization graphs on the web interface receive real-time updates as each piece of information processed by the MCU algorithm is transmitted via the USB port. It is worth mentioning that the figure provided is illustrative and represents one of the recorded signals during the experimental sessions.

Therapy analysis can be conducted concurrently with the pulse sensor during the session. The collected data are then made available on a web portal with various analytical options. Consequently, the assessment of autonomic therapy modulation using the developed device can be explored in future research involving a larger number of users for statistical validation.

The application of vagal stimulation therapy through this method has yielded promising results in the treatment of chronic diseases, showing efficacy comparable to traditional invasive approaches. Recent studies have underscored the significance of synchronizing vagal stimulation with the supervised respiratory cycle, particularly during the expiratory phase. In this context, the innovative device we have developed, coupled with a real-time signal processing algorithm, represents a major milestone in the field of vagal stimulation therapies. This device empowers the orchestration of stimulation sequences synchronized with the respiratory cycle over the course of treatment, facilitating closed-loop operation of the stimulator.

Furthermore, the designed firmware can accommodate other approaches, including the automation of variations in stimulation frequency parameters. This feature aims to mitigate the risk of users developing pharmacological tolerance to the treatment, which often necessitates higher stimulus dosages. This approach not only broadens the treatment options for a wide range of disorders but also enhances effectiveness and the overall patient experience. It offers a personalized and accessible approach to transcutaneous vagal stimulation (taVNS) treatment, ultimately promoting more humane and impactful outcomes.

## 4. Conclusions

In summary, this study presents the successful development of a feedback-driven stimulator prototype integrating real-time respiratory signal processing and autonomic assessment. Leveraging electronic design tools such as Eagle and 3D modeling via Catia V4, the device integrates a sophisticated algorithm embedded in the MiniEsp MCU for reliable detection of the expiration phase. The embedded software seamlessly interacts with the electronic circuit, allowing for the modulation of stimulation voltage pulse switching. The user-friendly visualization interface further enables flexible frequency selection, facilitating pulse delivery at speeds of up to 25 Hz. Offline signal analysis using the peak identification functions in Matlab demonstrated the algorithm’s robust performance, achieving an impressive 90% precision with slight delays of up to 150 ms. Nevertheless, the proximity of the sensor to oral and nasal regions may impact accuracy, particularly in the presence of coughing or speech. Through comprehensive operational and usability tests, the closed-loop device, coupled with peripherals, has demonstrated the potential for comprehensive stimulation, fostering optimism for future clinical validation and potential practical implementation.

## Figures and Tables

**Figure 1 healthcare-12-00031-f001:**
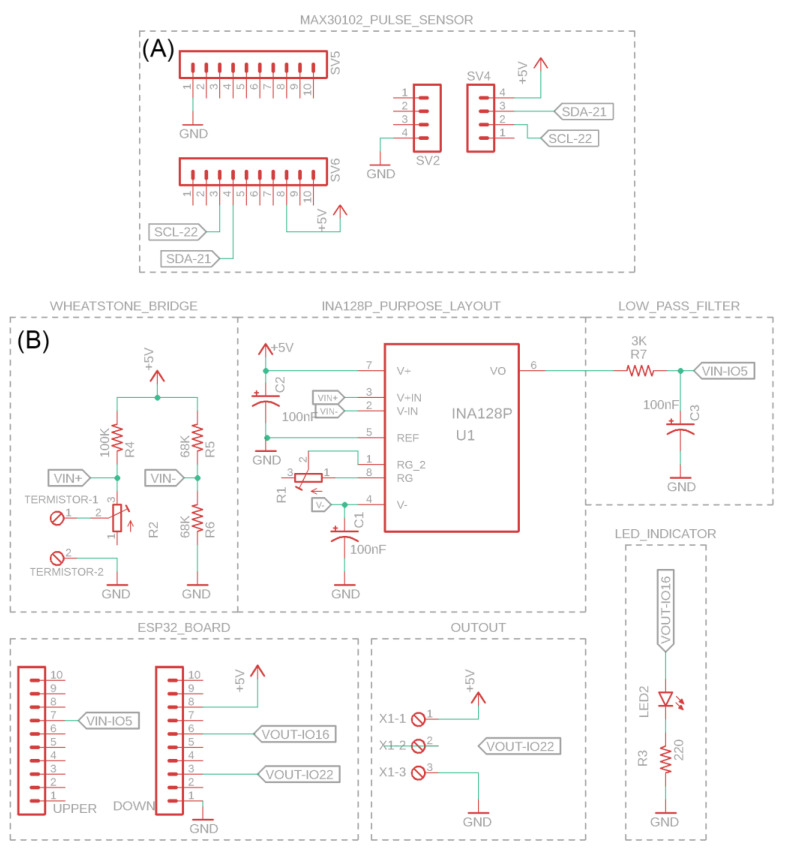
Construction of the logical diagram. (**A**) Electronic design for heartbeat measurement. (**B**) Electronic design for respiratory signal acquisition and conditioning module.

**Figure 2 healthcare-12-00031-f002:**
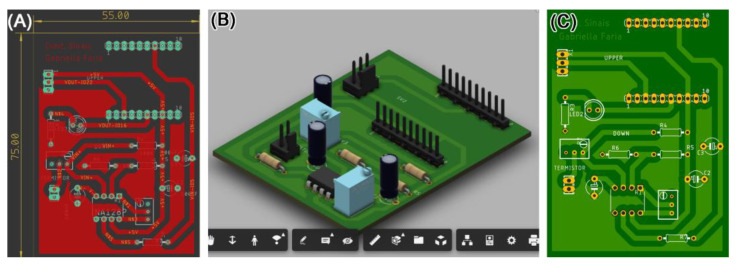
Electronic circuit design and printed circuit board. (**A**–**C**) Illustrate sequential steps within the Eagle software: (**A**) defines component placement, track design and the dimensions of the circuit board; (**B**) validates assembly through 3D component insertion; (**C**) generates the footprint for circuit board evaluation before printing.

**Figure 3 healthcare-12-00031-f003:**
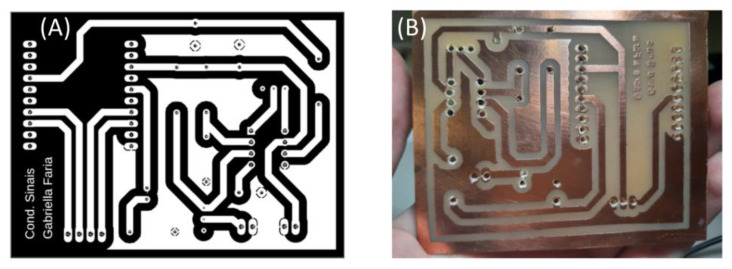
Printing the circuit for the photolithography process of the board. (**A**) Black and white printing model. (**B**) PCB board after the corrosion process, cleaned and drilled.

**Figure 4 healthcare-12-00031-f004:**
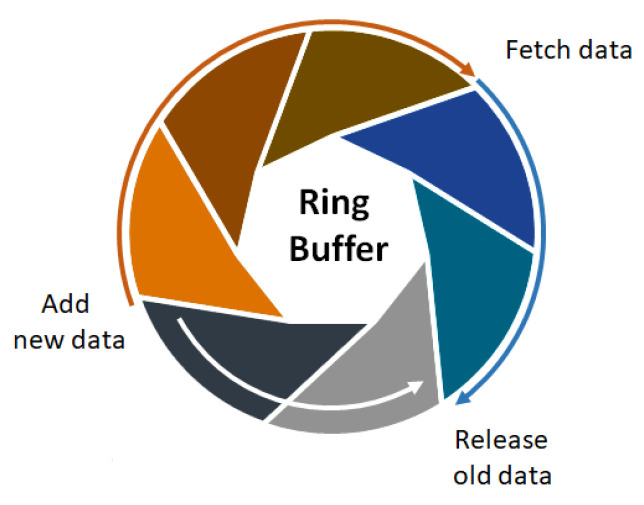
Schematic representation of the ring buffer. When new data are inserted and the vector reaches its capacity limit, the new point overwrites the oldest point at the beginning of the data structure.

**Figure 5 healthcare-12-00031-f005:**
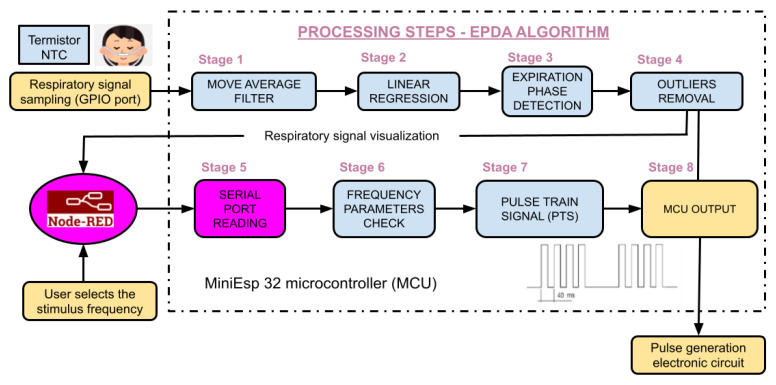
Schematic of the logical flow of the expiration phase detection algorithm and pulse train switching. The stimulation frequency is selected by the user.

**Figure 6 healthcare-12-00031-f006:**
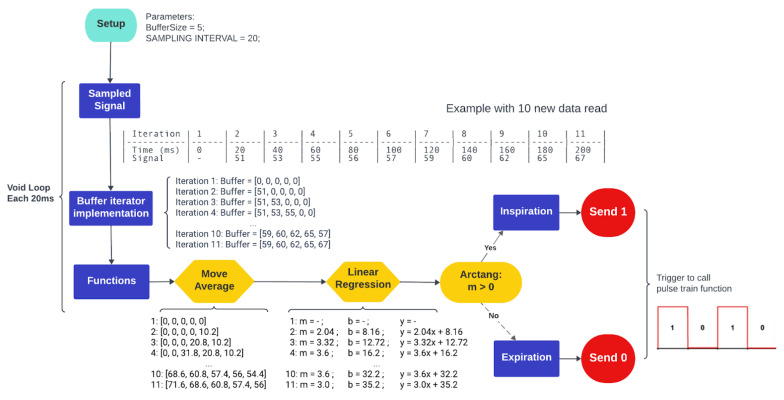
Diagram illustrating 11 iterations of the algorithm sequentially for each new sampled respiratory signal value.

**Figure 7 healthcare-12-00031-f007:**
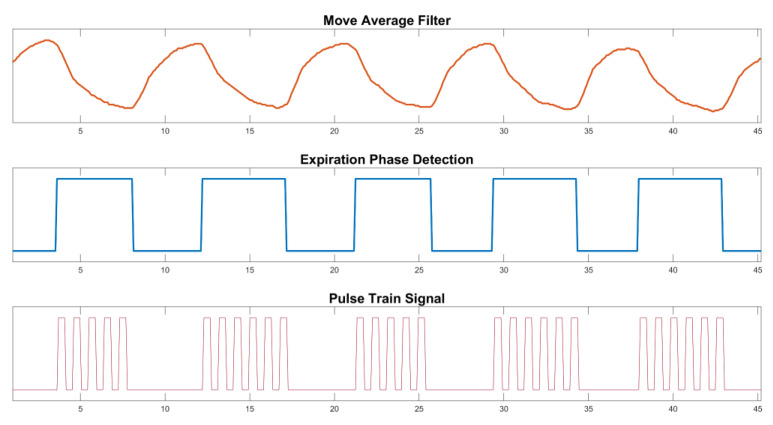
Results presented in accordance with the algorithmic stages: Moving average filter in step one; Rectangular pulse signal in step three; and Pulse Train Signal according to the frequency selected by the user in stage seven.

**Figure 8 healthcare-12-00031-f008:**
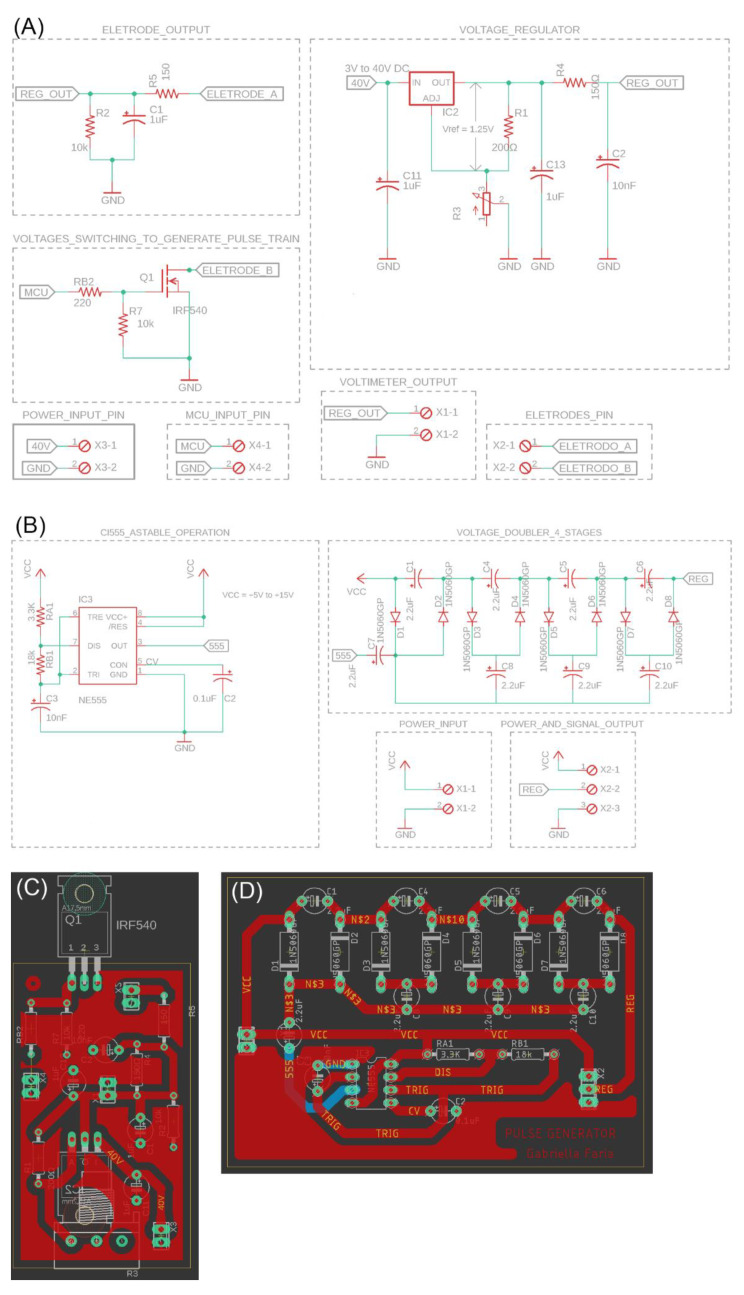
Logical diagram of second and third board of pulse generation circuit (PGC). (**A**) Logical diagram of voltage regulation module connected to the switching circuit. (**B**) Logical diagram of voltage doubler circuit. (**C**) Board project in Eagle software for (**A**). (**D**) Board project in Eagle software for (**B**).

**Figure 9 healthcare-12-00031-f009:**
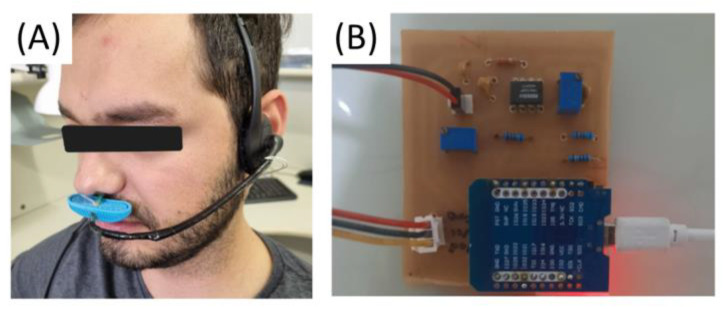
Respiratory signal acquisition and conditioning module. (**A**) Nasal thermistor positioned near the nostril. (**B**) Circuit components soldered, and MCU positioned.

**Figure 10 healthcare-12-00031-f010:**
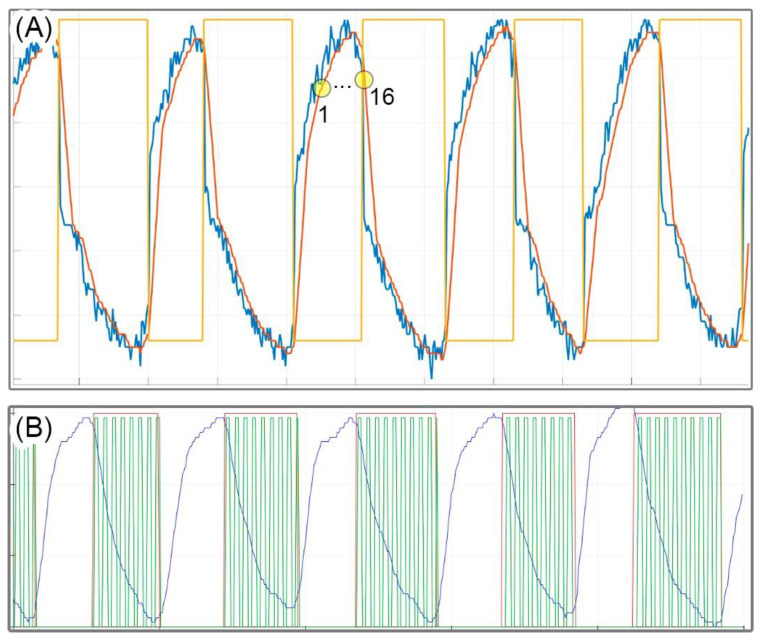
Results of the EPDA algorithm’s phases. (**A**) A rectangular pulse (orange signal), a digitally filtered respiratory signal (blue line), and a respiratory signal (red line). (**B**) A PTS signal associated with a particular frequency of 20 Hz is illustrated in conjunction with the breathing signal and square wave.

**Figure 11 healthcare-12-00031-f011:**
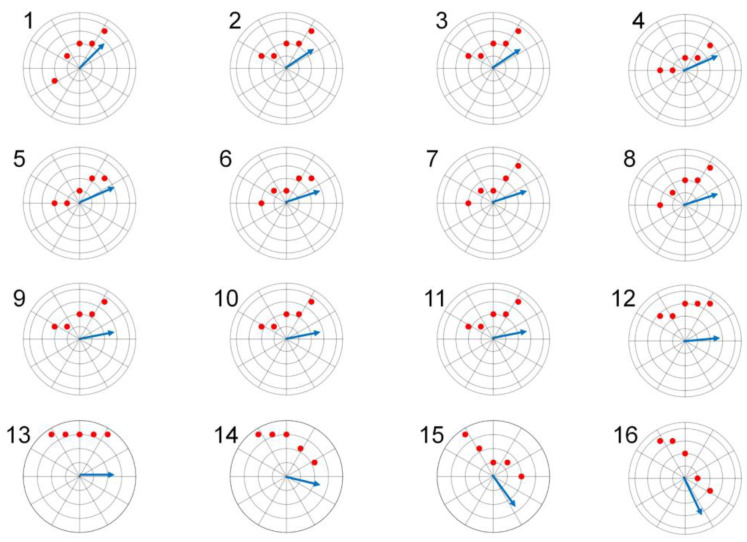
The angle evolution derived from linear regression is illustrated for select signal segments spanning from region 1 to region 16. In this diagram, the red points symbolize actual values sampled from the respiratory signal, while the blue arrow denotes the slope of the straight line derived from real-time linear regression calculations using these five sampled points.

**Figure 12 healthcare-12-00031-f012:**
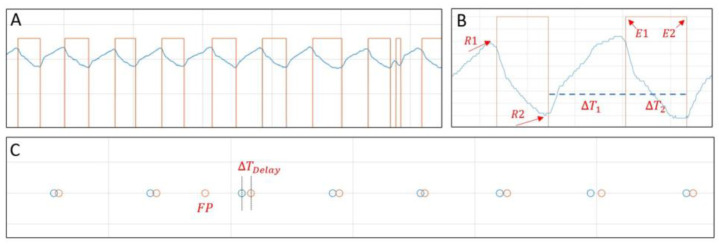
Respiratory signal and statistical analysis with real data. (**A**) Respiratory signal (blue signal) and rectangular pulse of EPDA output (red signal). (**B**) Mainly points for delay time analysis, where time between exhalation beginning chase (R1) and high level of EPDA signal is measured. (**C**) Time delay and false positive moments are indicated.

**Figure 13 healthcare-12-00031-f013:**
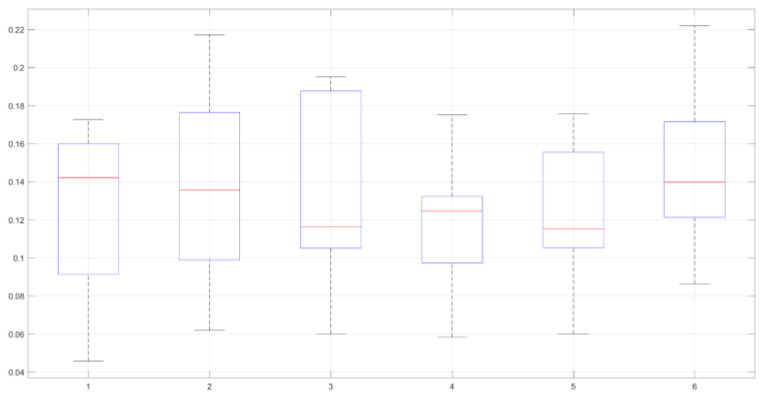
Boxplot analysis. Evaluation of response delays.

**Figure 14 healthcare-12-00031-f014:**
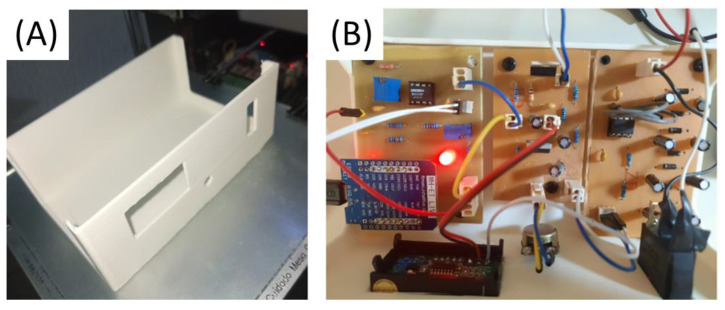
Case and boards: (**A**) 3D printing of case model for validation; (**B**) three circuit boards mounted in the case.

**Figure 15 healthcare-12-00031-f015:**
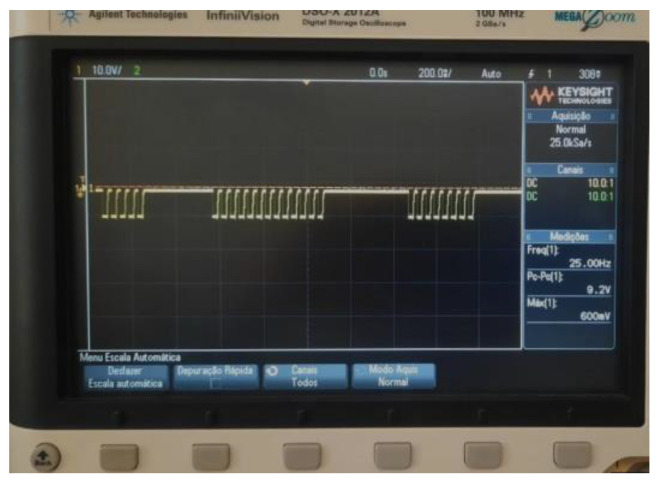
Signal generated by the PGC, synchronized by user’s breathing, visualized on the oscilloscope.

**Figure 16 healthcare-12-00031-f016:**
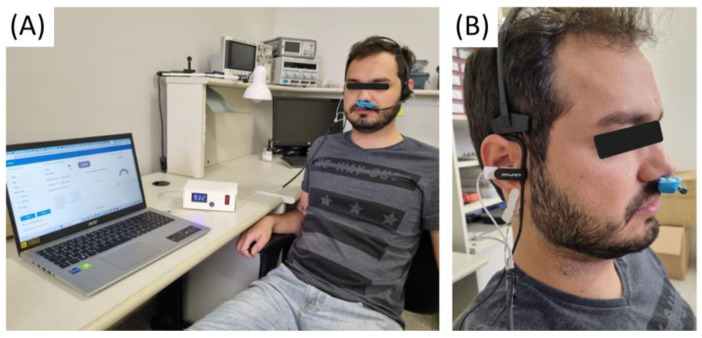
Stimulation device connected to the temperature sensor, stimulation electrodes, and a notebook for monitoring the therapy through the web interface. (**A**) Stimulator connected to the user’s peripherals. (**B**) Electrodes positioned in the auricular region and nasal thermistor positioned near the nostril.

**Figure 17 healthcare-12-00031-f017:**
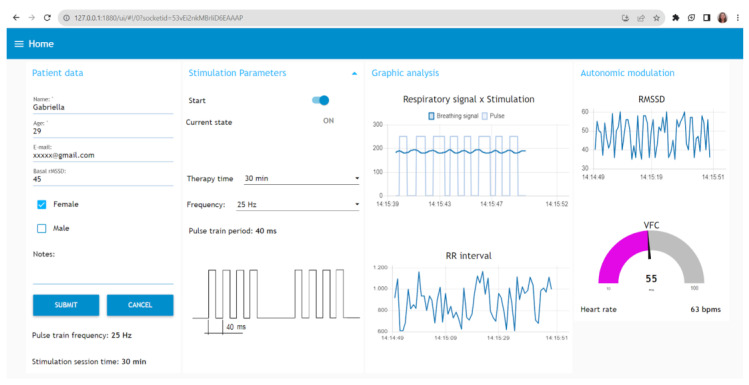
Web interface for patient data collection, visualization, and selection of stimulation parameters. Exemplary signals recorded during the experimental sessions.

## Data Availability

Data are contained within the article.

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
