# Peer review of "Advances in Non-Invasive Neuromodulation: Designing Closed-Loop Devices for Respiratory-Controlled Transcutaneous Vagus Nerve Stimulation"

_healthcare, 2023, doi:10.3390/healthcare12010031_

Round 1

Reviewer 1 Report

Comments and Suggestions for Authors

1. Overall the scientific approach of this research/paper is technically sound. Hardware design was presented and experimental data was presented to support the data collection and analysis. 

2. If the circuit diagram of the entire system could be added, it would be easier for readers to understand the overall design of the system.

3. Stimulatione in the title is wrong

4. Università degli Studi di Milano, Milan, ItalyAffiliation, remove affiliation;

5. wrong font size line 365,

6. For online algorithm, I feel the only way to improve the response time and reduce transport delay is by not collecting all the data instead only using a moving window to analyze and extract rising and falling signals in real time. only have to record the during of inspiration and expiration for plotting and statistical analysis.  Essentially, the real data shown in Figure 7 may not need to be collected, instead only the postprocessed square wave is needed. That should be much faster and much less storage space.

7. Overall the data driven approach is encouraging and quite convincing. 

Comments on the Quality of English Language

definitely need some editing and proof reading. It appears that the paper was rushed somehow. 

Author Response

RESPONSES TO REVIEWERS

Author's Reply to the Review Report (Reviewer 1)

We greatly appreciate your time in reviewing this manuscript. The detailed responses can be found below. The necessary corrections have been emphasized in the revised article. Yellow-highlighted text represents newly added or modified points, while blue-highlighted text signifies enhancements to the language.

Review 1: If the circuit diagram of the entire system could be added, it would be easier for readers to understand the overall design of the system.

Response: We appreciate your suggestion. In response to your comment, we have included methodology section circuit diagrams, electronic schematics, and printed circuit board layouts for each of the three physical modules present in the developed stimulator device: Signal acquisition and conditioning module, Pulse Generation Circuit (PGC), and switching circuit to generate Pulse Train. We believe these additions will significantly enhance the reader's understanding of the overall system design.

Review 1: Stimulatione in the title is wrong. Università degli Studi di Milano, Milan, ItalyAffiliation, remove affiliation;  wrong font size line 365,

Response: We appreciate your feedback and sincerely apologize for any errors. The revised version now incorporates the necessary corrections.

Review 1:  For online algorithm, I feel the only way to improve the response time and reduce transport delay is by not collecting all the data instead only using a moving window to analyze and extract rising and falling signals in real time. only have to record the during of inspiration and expiration for plotting and statistical analysis.  Essentially, the real data shown in Figure 7 may not need to be collected, instead only the post processed square wave is needed. That should be much faster and much less storage space.

Response:  Thank you for your suggestion. Indeed, it is not necessary to sample with such precision as illustrated in Figure 7. The inclusion of the figure was primarily intended to show the reader the collected signal. Based on this insightful observation, we have made adjustments to the methodological description concerning this step.

We have provided additional details in Section 2.2, "Real-time implementation of the physiological signal processing respiratory algorithm" within the Methodology, as well as in the associated results. These modifications clarify that only the duration of inspiration and expiration is recorded for real-time analysis, reducing the data size and improving response time. 

Review 1: Overall the data driven approach is encouraging and quite convincing. 

Response: We appreciate your positive feedback on our data-driven approach.

Review 1:   Comments on the Quality of English Language definitely need some editing and proof reading. It appears that the paper was rushed somehow.

Response: We appreciate your input regarding the English language usage in our manuscript. Consequently, we have diligently undertaken comprehensive editing and proofreading to enhance linguistic accuracy.

Reviewer 2 Report

Comments and Suggestions for Authors

- The merit of this study is not clearly shown. 

- The manuscript lacks several key and complete details on both design and experimental aspects. 

- The design and complete circuits are not clearly and completely given. 

- The computational algorithms/methods used in the microcontroller are not clearly and completely shown. 

- The performance on signal processing algorithm is not individually examined and reported in the manuscript. 

- Also, the results on the pulse generation and the stimulation are not shown. 

- The number of cases, subjects, their demographic information, and experimental trials on using the developed device are not clearly reported. 

- From the manuscript, it thus implies that the developed device and its sub-systems are not completely tested. 

- The title does not properly portray the content of manuscript. 

Comments on the Quality of English Language

- There are several places that need to be corrected. 

- In addition, the terms or phrases used need to be proper and precise; for example, real-time implementation. 

Reviewer 3 Report

Comments and Suggestions for Authors

This manuscript presents the development of an innovative closed-loop device for non-invasive transcutaneous auricular vagus nerve stimulation (taVNS), designed to treat conditions such as epilepsy and depression. The device features real-time signal processing that aligns electrical stimulation with the patient's expiratory phase, which could enhance neuromodulation effectiveness. Equipped with sensors for physiological data collection, software for management and analysis, and demonstrating a 90% accuracy rate in respiratory peak detection with minimal delay, the prototype shows promise for clinical application pending further refinement and trials. The manuscript is well-structured and coherent. The originality of the work is significant, and it is a suitable candidate for publication, subject to the following revisions:

  1. The text references a Figure 9C on Line 339; however, Figure 9 appears to lack a panel C. Please address this discrepancy.
  2. In Line 445, regarding disclosures of interest for patents, the authors should state 'No conflict of interest' to align with common academic standards.
  3. The supplementary materials mentioned on Line 447 are not included. Please provide these materials to facilitate the review process

Round 2

Reviewer 2 Report

Comments and Suggestions for Authors

- The revised manuscript was improved from the original manuscript.

- However, it still lacks the clear and complete validation on each component implemented and composed into the complete device. The results shown generally exhibit that the developed device is functioning. 

- In addition, the algorithms were not clearly presented and expressed in mathematical form. There are not any discussion regarding on the design and parameters selection. 

Comments on the Quality of English Language

There are some terms that are unconventional and not precise. 
